# Asthma and Tobacco Smoking

**DOI:** 10.3390/jpm12081231

**Published:** 2022-07-27

**Authors:** Vanesa Bellou, Athena Gogali, Konstantinos Kostikas

**Affiliations:** 1Department of Respiratory Medicine, School of Medicine, University of Ioannina, 45110 Ioannina, Greece; athenagogali@yahoo.com (A.G.); ktkostikas@uoi.gr (K.K.); 2Department of Hygiene and Epidemiology, School of Medicine, University of Ioannina, 45110 Ioannina, Greece

**Keywords:** asthma, smoking, tobacco, exacerbations

## Abstract

Asthma is a prevalent chronic pulmonary condition with significant morbidity and mortality. Tobacco smoking is implicated in asthma pathophysiology, diagnosis, prognosis and treatment. Smokers display increased prevalence and incidence of asthma, but a causal association cannot be claimed using existing evidence. Second-hand smoking and passive exposure to tobacco in utero and early life have also been linked with asthma development. Currently, approximately one-fourth of asthma patients are smokers. Regular smokers with asthma might display accelerated lung function decline and non-reversible airflow limitation, making their distinction from chronic obstructive pulmonary disease patients challenging. Asthma patients who smoke typically have uncontrolled disease, as shown by increased symptoms, more exacerbations and impaired quality of life. On the other hand, smoking cessation improves lung function and asthma severity. Thus, asthma patients and their caregivers should be actively questioned about their smoking status at each medical encounter, and smoking cessation ought to be strongly encouraged both for patients with asthma and their close contacts. Smokers with asthma should be provided with comprehensive smoking cessation interventions on top of other anti-asthma medications.

## 1. Introduction

Asthma is a common chronic airways disease characterized by variable expiratory airflow obstruction [1]. The latest Global Burden of Disease estimated the global prevalence of asthma at 262 million cases (95% UI, 224–309), based on available evidence from 2019 [2]. The trademarks of asthma are airway inflammation and airway hyperresponsiveness, manifesting with varying degrees of dyspnea, wheezing, cough and/or chest tightness. Several environmental factors, including tobacco, might trigger exacerbations of the disease, i.e., episodes in which symptoms worsen to an extent that warrants modification of a patient’s treatment [1,3].

Smoking is a hazardous health habit associated with significant morbidity and mortality population wide [4,5]. According to the latest Global Burden of Disease, tobacco is the second leading risk factor of death and the third leading risk factor of Disability-Adjusted Life Years (DALYs) worldwide, accounting for 8.71 million (95% UI, 8.12–9.31) deaths and 230 million (95% UI, 213–246) DALYs in 2019, respectively [5]. These effects are a consequence of the association of tobacco, with increased risk of a multitude of chronic diseases, as well as infections and acute health conditions [4,6]. 

Smoking has multiple ramifications for respiratory health [7]. The present review focused on the various links between smoking and asthma, as supported by existing literature.

## 2. Methods

We performed a review of the literature across two databases (PubMed and Google Scholar), using relevant keywords: asthma and tobacco, smoking or smoke. We searched for systematic reviews, meta-analyses, observational studies and clinical trials that examined the effect of various exposures to tobacco on asthma. We included all the following types of studies: studies that examined smoking as a risk factor or a prognostic factor of asthma, studies that estimated the epidemiology of smoking in patients with asthma, studies that examined diagnostic and/or therapeutic deviations in asthmatic smokers and studies that assessed the efficacy and/or efficiency of smoking cessation options among asthma patients. We excluded studies that did not involve humans and were not published in the English language. We did not enforce any limitation on the date of publication.

## 3. Smoking as a Risk Factor of Asthma

Current evidence suggests that asthma is not the result of a single environmental or genetic cause, but develops due to interplay between multiple genetic and environmental factors. Moreover, asthma is quite a heterogeneous disease, which is commonly categorized in phenotypes, i.e., subgroups of patients that have distinct clinical manifestations. These phenotypes are considered the output of different pathophysiological processes, suggesting that asthma is a complex disease with many mechanisms contributing to disease etiology and natural course [8,9]. A variety of risk factors and environmental exposures have been linked to asthma development, either increasing or decreasing the risk of the disease. These include demographic factors, such as age and sex; developmental factors, such as preterm birth; mode of delivery and history of infections; socioeconomic status-related factors, such as agricultural subsistence, income and daycare enrollment; dietary factors and medications; and, last but not least, inhaled exposures, such as tobacco, air pollution and air allergens [9]. The latest evidence from Global Burden of Disease placed smoking second among the leading risk factors for DALYs attributed to asthma [2]. 

There are a multitude of observational studies that have depicted increased prevalence of asthma in cigarette smokers [7,10,11,12,13,14,15]. Cohort studies have also shown increased incidence of asthma among cigarette smokers [7,15,16,17]. To elaborate on the association between tobacco exposure and asthma, secondhand exposure to smoking should also be considered. A few observational studies have shown increased frequency of passive smoking among patients with asthma, indicating that secondhand smoking is also a risk factor of asthma [7,18,19]. Moreover, there is emerging evidence depicting an association between the use of electronic cigarettes and asthma, chronic obstructive pulmonary disease (COPD) and their coexistence, often mentioned as asthma–COPD overlap (ACO) [20,21]. This association persisted even after controlling for tobacco smoking and other disease-relevant factors [22]. To complicate matters even more, there are data supporting that asthma is a risk factor of COPD and that the association persists even after adjusting for smoking [23].

While tobacco smoking is considered an established risk factor of asthma, a causal association cannot be claimed. The evidence supporting the association of smoking with asthma is comprised of observational studies, with many of them being cross-sectional and case-control studies [18]. These studies are prone to various biases such as confounding, reverse causality and selection bias; therefore, the study design should be examined carefully and the results interpreted with caution [24,25]. Published Mendelian Randomization studies examining a potential causal relationship between smoking and asthma had conflicting results and did not shed further light [26,27]. 

One of the Bradford Hill criteria required to determine if an observed association is causal is a temporal sequence, i.e., the exposure should precede the outcome [28]. It should be noted that the majority of asthma cases are diagnosed in childhood; thus, active smoking and continuous passive exposure to smoking are not likely causal factors of childhood asthma [7,8,18,29]. However, it might be the case that smoking, whether active or passive, is among the exposures causing asthma of a later onset in adult life [7,18,29,30]. The fact that asthma onset typically occurs in childhood has shifted the search for risk factors occurring in prenatal, perinatal and early life [31]. Thus, one factor that has received attention is exposure to tobacco during the years preceding childhood, which are critical for lung development [9,32]. 

Passive exposure to tobacco in utero and/or early life has been linked with various adverse effects on the respiratory system across the life span of exposed individuals. To elaborate, exposure to maternal smoking has been associated with increased risk of respiratory infections in childhood, wheezing, as well as diminished lung function in childhood and adolescence [32,33]. Epidemiological studies have shown that the effect on lung function persists in adult life and people exposed to maternal smoking also have higher incidences of airflow obstruction, chronic obstructive pulmonary disease and idiopathic pulmonary fibrosis [34,35,36,37]. Regarding asthma, both maternal and paternal smoking have been associated with the development of asthma in offspring [7,38]. Recently published transgenerational studies have also depicted increased risk of asthma in children whose grandparents smoked [39,40]. Alas, more research is needed to verify this “vertical transmission” of smoking-induced asthma risk and elucidate the mechanisms behind it [41].

Taking into consideration that asthma remains a disease without a cure, public health efforts have focused on primary prevention measures to tackle its morbid impact. Therefore, population-wide measures that aim to lessen the incidence of smoking have been proposed as preventive measures to lessen the incidence of asthma in descendants of potential smokers [9,31].

## 4. Diagnostic Challenges for Asthmatic Smokers

Prevalence rate estimates the smoking range from 20% to 35% of asthma patients in published population-wide studies, regardless of the country of origin of the sample used. The prevalence of active smoking in patients with asthma is, therefore, approximately the same as the prevalence in the general population [18,42,43,44,45,46]. In spite of the high prevalence of smoking among patients with asthma, this group of patients is typically excluded from randomized clinical trials assessing the effectiveness of various inhaled medications in asthma patients [18,47]. Active smokers and/or patients with a smoking history of over 10 pack years were also excluded in the majority of recent randomized clinical trials conducted to examine the efficacy of biologics in severe asthma patients [48].

The main symptoms that trigger a diagnostic examination for asthma are shortness of breath, cough, chest tightness and wheezing. Typically, patients have more than one of these symptoms, and symptoms have a certain pattern: they vary over time and in intensity, tend to worsen at night and early morning and may be triggered by certain exposures, such as tobacco and allergens [1,8]. An important clinical diagnostic challenge is to distinguish between the two most prevalent airflow limitation disorders, asthma and COPD. COPD is a common chronic respiratory condition characterized by persistent symptoms and airflow limitation due to airway and/ or alveolar abnormalities. It has different pathophysiology than asthma and is typically diagnosed in midlife or old age [49,50]. The main symptoms of COPD are similar to the ones of asthma, but lack the characteristic variable pattern displayed in asthma [8,50]. Specifically, dyspnea presents on exertion in the early stages of COPD, while in asthma it is present during exposure to triggers, at night and/or during exacerbations, and asthma patients are more likely to report wheezing and less likely to have chronic bronchitis symptoms [51,52,53]. Other attributes that might be helpful for differentiating asthma from COPD include sex, information from the patient’s family and personal history (family history of asthma, allergen sensitization, history of hay fever, eczema or allergic rhinitis, presence of comorbid diseases such as cardiovascular disease) and biomarkers (eosinophils, IgE, FeNO) [52,53]. A helpful diagnostic algorithm, adapted from the Greek Guidelines for COPD, is presented in Figure 1 [54]. Last but not least, in the early stages of COPD, clinical examination is normal, as it is in asthma [8,50].

The diagnosis of asthma is made via lung function testing in patients with a clinically relevant presentation. Spirometry in asthma displays expiratory airflow limitation and excessive variability of lung function [1,8]. Existing literature has depicted that smoking initiation for the majority of people happens at a young age, and the transition to regular smoking happens between adolescence and young adulthood [55]. Nicotine dependence develops very quickly in young people with symptoms of asthma, making quitting a difficult process [56]. Asthma patients that have regularly smoked since childhood or early adulthood are more prone to impaired lung function and airway remodeling, ultimately causing fixed (non-reversible) airway obstruction [57,58,59,60,61]. This makes differentiation of asthma patients with a smoking history from COPD patients a strenuous task, given that smoking is the main risk factor of COPD and COPD patients have poorly reversible airflow limitation [50]. On top of this, certain COPD patients manifest airway hyperresponsiveness, which is the trademark feature of asthma [62]. However, the combination of greater lung function values, slower rates of lung function decline and marked airway hyperresponsiveness should steer the clinician towards an asthma diagnosis [52,53].

There are some patients that have characteristics of both asthma and COPD. This entity is better known as asthma COPD overlap (ACO), but its presence and characteristics have raised controversies among various professional societies and clinical experts. There are multiple different definitions of the disease, some of which require a known history of asthma, while others require a smoking history among other diagnostic criteria [63]. Patients with ACO display a different prognosis and need different treatment than patients with asthma and COPD, and thus should be sought after and handled accordingly. The most important therapeutic intervention in current or ex-smokers with COPD who present an asthmatic component is they should not miss the benefits of ICS in the appropriate dosing, especially in the presence of eosinophilic/T2 inflammation [64]. Existing guidelines on ACO have been revised extensively and repeatedly over the past few years to incorporate accumulating new evidence, and hopefully a consensus will be reached in the coming years [63,65].

## 5. Prognostic and Treatment Implications in Asthmatic Smokers

Smoking is implicated in the prognosis of asthmatic patients in a number of ways, with consequences such as increased morbidity and mortality [18,66]. These phenomena could be attributed to the modifications made by tobacco in airway morphology and the inflammatory processes of asthma [18,42]. Active smokers with asthma present increased neutrophils in induced sputum and reduced pH and squamous cell metaplasia [67]. Past smokers with asthma present airway autoimmunity and increased eosinophilic inflammation and activation, with reduced sensitivity to corticosteroids [68]. Cigarette smoking also suppresses FeNo, which is a helpful marker of asthma activity. Additionally, patients with asthma who smoke may present an accelerated rate of decline in lung function and may develop persistent airflow obstruction due to airway remodeling [57,58,59,60]. Asthmatic smokers might have aggravated small airway obstruction and an altered microbiome, with greater bacterial diversity [69,70,71]. The specific effects of smoking on distinct asthma phenotypes have not received much attention, so more research is needed to clarify the effect of smoking on different phenotypes of asthma [18,42].

Active smokers with asthma have an increased burden of symptoms, both intensity wise and frequency wise, as measured using suggested questionnaires in published studies [72,73]. Smokers with asthma exhibit higher absenteeism from work and school; have increased use of rescue medication, which is a proxy for more symptoms and inadequate disease control; and worse indices of health status compared to non-smokers [74,75]. Asthma patients who smoke have an elevated rate and severity of exacerbations [66]. Specifically, more than half of current smokers had at least one exacerbation per year requiring systemic corticosteroids, compared to 40% of former smokers [76]. Previous literature has also shown that long-term smoking directly increases the need for healthcare use in asthma patients, in the form of visits to the emergency department, visits to the general practitioner or unscheduled appointments with the pulmonologist [74,77,78]. Furthermore, smoking is linked with increased hospitalizations in asthmatic patients and asthma-related deaths [66]. Furthermore, there is scant evidence that electronic cigarette use and second-hand exposure to their aerosols may also amplify symptoms and increase exacerbations in asthma patients [79,80,81].

Furthermore, asthmatic smokers display greater prevalence of various comorbid conditions, according to published studies. These conditions include perennial rhinitis, seasonal rhinitis, lung cancer, coronary heart disease, arrhythmias, hypertension, diabetes mellitus, osteoporosis and prostate hyperplasia [82,83]. The presence of any of these comorbidities in a patient with asthma influences his overall health status, as well as the course of asthma, by means of directly damaging asthma control and potential drug interactions [83]. Taking into account all of the aforementioned information, it does not come as a surprise that ever smokers with asthma have a worse quality of life than never smokers with asthma [18].

Active smokers with asthma are less likely to adhere to proper treatment and less prone to follow asthma education programs [75]. At the same time, smoking interferes with treatment modalities received by patients with asthma because the efficacy of certain medications is altered by the smoking status of a patient. A multitude of clinical studies have shown that smokers have relative resistance to inhaled corticosteroids, which is the mainstay controller treatment option in asthma patients, according to the latest guidelines [19,84,85]. Moreover, smokers have reduced sensitivity to leukotriene receptor antagonists [85,86]. A therapeutic option to reduce exacerbations in difficult-to-treat uncontrolled asthma patients with T2 low phenotype is long-term azithromycin; however, it is effective only in non-smokers [87,88].

Combining the negative effects of smoking on asthma symptoms and exacerbations, which are the two aspects assessed to rate the control and the severity of the disease, it becomes clear that asthmatic smokers usually have more severe and uncontrolled disease and thus display a need to intensify treatment. A small number of real-life trials have shown that the main treatment options used in asthma patients remain effective in smokers, as shown in Table 1 [89,90]. ICS is the mainstay controller medication in asthmatic smokers, but patients might require higher doses due to the relative resistance they display. Fine and extra-fine particle ICS might also have an advantage over other molecules in asthmatic smokers to better target small airways [90,91,92,93]. Smokers with asthma might also benefit from earlier introduction of long-acting beta agonists or long-acting anti-cholinergics [18,42,47]. More pragmatic clinical trials, with less strict exclusion criteria, should be conducted to elucidate the comparative efficacy of various medications and treatment strategies in asthma patients [18,42,94].

Second-hand smoking ought not to be forgotten, as it impacts asthma prognosis in similar manners to active smoking. Asthma patients exposed to passive smoking present graver symptoms and more severe disease, worse health outcomes, impaired health status and quality of life and, last but not least, more exacerbations. These findings were consistent in all asthma patients regardless of age (in either children or adults) [110,111]. The harmful effects of tobacco on asthma disease control might be aggravated in patients that are concurrently exposed to indoor or outdoor air pollution. These effects include asthma control, severity of disease and lung function. Potential mechanisms of this effect include the exacerbation of inflammation and allergen-induced inflammation due to Th2 responses [112,113].

## 6. Smoking Cessation in Asthma Patients

The realization that asthma constitutes a syndrome rather than a single disease has created a paradigm shift in its treatment in recent years [9]. This is mainly applied to patients on the severe spectrum of the disease, i.e., patients with poor symptom control and/or frequent exacerbations while receiving maximal optimized controller therapy [48,114]. For these patients, instead of enforcing a “one size fits all” strategy, therapy is personalized depending on disease phenotype and certain patient characteristics [48]. Considering previous discussions on the various adverse effects of tobacco on the prognosis of asthma, smoking cessation is strongly encouraged by the latest guidelines on asthma management. Furthermore, tobacco smoking is nowadays considered among comorbid diseases as a treatable trait that should be targeted to improve asthma control [114,115]. Both the Global Initiative of Asthma and joint European Respiratory Society/American Thoracic Society guidelines cite smoking as a potential contributory factor of severe asthma, i.e., asthma that requires treatment with high doses of ICS, plus a second controller to achieve disease control or that remains uncontrolled [116,117].

It is established that the benefits of smoking cessation begin right after quitting and carry on throughout a person’s life. It should be stressed that smoking cessation impacts asthma patients’ overall health and physical status, as well as their asthma prognosis [4]. Published literature has illustrated various beneficial effects of smoking cessation on the prognosis of patients with asthma. In the short term, smoking cessation leads to a reduction in symptoms and less frequent use of rescue medication. In the long term, smoking cessation in asthmatics leads to improved lung function and a better quality of life. Moreover, asthma patients who quit require a smaller dose of inhaled corticosteroids to control their disease. Similar benefits of smaller magnitude have been observed for active smokers with asthma who reduced the number of cigarettes they consumed [118,119,120]. Mechanistically, smoking cessation is hypothesized to alter the inflammatory phenotype of asthmatics, which is manifested by a reduction in the number of sputum neutrophils and an increase in FeNO [118,119]. Quitting is also linked with improved airway hyperresponsiveness, as displayed by methacholine and histamine provocation testing [119,120].

Several smokers attempt to quit on their own, but quitting smoking without any aid has very low abstinence rates [121]. There is consensus that the most efficient smoking cessation intervention is a comprehensive treatment, combining behavioral counseling, pharmacotherapy and follow-up support [4,122]. Currently, there are seven effective pharmacological treatments for smoking cessation approved by the US Food and Drug Administration. These include varenicline, bupropion and five forms of nicotine replacement therapy [4,122]. The efficacy of abstinence for these options, when compared to placebo, are estimated at RR 2.24 (95% CI, 2.06–2.43) for varenicline, 1.64 (95% CI, 1.52–1.77) for bupropion, 1.49 (95% CI, 1.40–1.60) for gum, 1.64 (95% CI, 1.53–1.75) for patches, 1.52 (95% CI, 1.32–1.74) for lozenges, 1.90 (95% CI, 1.36–2.67) for inhalers and 2.02 (95% CI, 1.49–2.73) for nasal spray. The most common side effects for each of the above medications are: nausea for varenicline, insomnia for bupropion, jaw pain for gum, skin reactions for patches, hiccups for lozenges, cough for inhalers and nasal irritation for nasal spray [4,122].

The number of clinical trials examining the efficacy of various smoking cessation strategies and medication, exclusively in asthma patients, is very limited [120,123,124,125]. More clinical trials should be designed to assess and compare the effectiveness of existing smoking cessation interventions in asthma patients. Existing guidelines for asthma do not make any specific recommendations for smoking cessation interventions (i.e., specific therapeutic options and elaborate regimens or doses), while they emphasize that quitting should be encouraged and cessation support should be offered for all patients [18,75]. Published literature does not support the use of electronic cigarettes as a method to encourage smoking cessation, neither in the general population nor in asthma patients specifically [81,122].

The latest guidelines recommend that all patients should be questioned about their smoking status at every encounter, as if it were a vital sign, and suggest the use of 5As to screen for smoking in clinical settings [4,122]. Hospitalizations for an exacerbation in asthma patients could serve as an opportunity to screen for active or passive smoking. Active smokers could be provided with smoking cessation interventions, or at least be offered simple advice, and then be referred to treatment resources, such as telephone lines, websites or specialty treatment programs. This opportunity is so far underutilized in clinical practice [4,126]. Passive exposure to tobacco should not be dismissed. Adult smokers should be advised against secondhand exposure to tobacco. It is also imperative to advise parents and caregivers of asthmatic children to quit smoking, as well as to assist expectant mothers to quit [122,127,128].

## 7. Conclusions

Asthma and smoking intertwine in various ways (Table 2). Both active and passive smoking are commonly considered risk factors of asthma, but evidence of a causal association is conflicting. Despite the well-known hazardous effects of tobacco on respiratory health, smoking is a quite common habit among asthma patients. Smoking affects airway morphology and alters inflammation of asthmatic individuals, worsening disease prognosis and making the differential diagnosis of asthma and COPD rather laborious. Quitting smoking improves both the lung function and symptoms of asthma patients, rendering receipt of comprehensive smoking cessation interventions imperative for proper treatment.

## Figures and Tables

**Figure 1 jpm-12-01231-f001:**
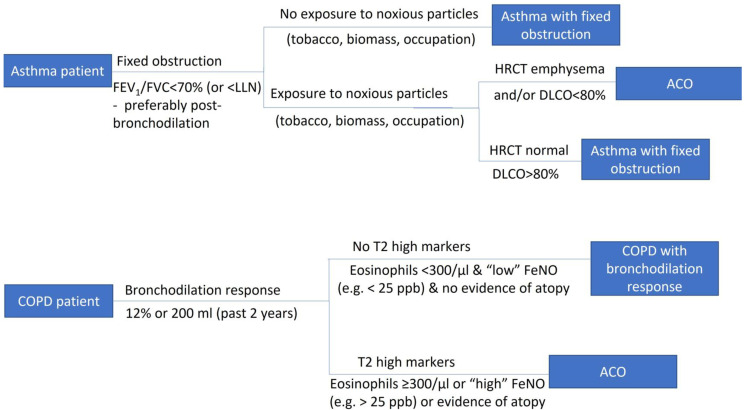
Diagnostic algorithm for asthmatic smokers.

**Table 1 jpm-12-01231-t001:** Controller treatment options for asthma patients with a smoking history, according to pertinent literature.

Medication	Findings from Clinical Studies
ICS	ICS remain effective in asthma patients with a smoking history, but some patients might have a blunted response to these treatments [19,95,96,97]. Higher ICS doses might be required to achieve asthma control in smokers [97]. Extrafine formulations might confer greater advantages, according to clinical data [98]
LABA	Adding LABA to ICS increased airway hyperresponsiveness and airway caliber and improved asthma symptoms in asthmatic smokers when compared to higher ICS doses [96,99].
LAMA	Tiotropium is an established treatment option for asthma; however, active smokers were excluded in tiotropium trials.The benefits of tiotropium in asthma control and lung function were marginally higher in ex-smokers with asthma [100] and tiotropium has shown effectiveness in patients with concomitant asthma and COPD [101]. Additionally, triple therapy (ICS + LABA + LAMA) improved small airway outcomes in asthmatic smokers [102].
OCS	Cigarette smoking diminishes the therapeutic response to OCS [103].
Macrolides	Azithromycin did not improve lung function and symptoms in a clinical trial of active smokers with asthma [87].
LTRA	Certain asthmatic smokers show marked improvement in asthma control after adding montelukast to ICS [86,104].
Theophylline	Combination of low-dose theophylline with ICS improved lung function and symptoms in asthmatic smokers [105].
Biologics	Anti-IgE is effective in active and ex-smokers [106,107]. Anti-IL5 and anti-IL5R are effective irrespective of smoking status [107,108,109]. Anti-IL4 has not been evaluated in active smokers or past smokers with over 10 pack years [48].

Abbreviations: COPD: Chronic Obstructive Pulmonary Disease; ICS: Inhaled Cortico-Steroids; LABA: Long-Acting Beta-Agonists; LAMA: Long-Acting Muscarinic Antago-nists; LTRA: LeukoTriene Receptor Antagonists; OCS: Oral CorticoSteroids.

**Table 2 jpm-12-01231-t002:** Summary of myths and realities of association of asthma and smoking.

Myth	Reality
Smoking causes asthma	There is no established causal association between tobacco exposure and asthma, and existing evidence is conflicting
Smoking hinders asthma diagnosis	Smoking may cause fixed airway obstruction, but other disease characteristics enable accurate diagnosis
Smoking impairs asthma prognosis	Smoking is associated with worse clinical outcomes and disease control in asthma patients
Smoking interferes in asthma treatment	The main anti-asthmatic medications remain effective in asthmatic smokers. However, there is some evidence for reduced efficacy of inhaled corticosteroids.

## Data Availability

Not applicable.

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
