# Peer review of "Asthma and Tobacco Smoking"

_jpm, 2022, doi:10.3390/jpm12081231_

Round 1
Reviewer 1 Report
Dear Authors, I have read the text with interest especially because it is highly clinically relevant. I do agree with authors general approach that smoking is harmful for asthmatic patients and comprehensive efforts should be undertaken decrease smoking burden especially in asthmatic. Although I find the review well written and useful especially for young physicians there are some aspects which in my opinion are crucial before acceptance of the manuscript for publication.
1. 1 Introduction
I would expect that the authors would describe in this part what was the methodology for review performance (systemic review/ random autonomic authors decision?). Which databases have been assessed? Which articles have been included or excluded, and what kea words have been used. In case the review was not systematic this should have been underlined in the introduction, but the method of literature search should be described.
2. 2 Diagnostic Challenges for Asthmatic Smokers: The authors explain most important challenges of differential diagnosis of asthma and COPD which includes Spirometry tests. Unfortunately dyspnea characteristics, which is weary different in asthmatic and COPD patients. This should be explained in detail with underlining nighttime dyspnea exacerbations are characteristic of asthma and exercise induced dyspnea in COPD.
3. 3 I do luck a distinct chapter in which tips for differential diagnosis of “patients with asthma who smoke with present an accelerated rate of decline in lung function and persistent airflow obstruction due to airway remodeling with COPD which has developed in allergic patient. (the FEV1 decline and bronchial reactivity in both diseases should be addressed in larger detail.
4 Most importantly the Review in its current form is not addressing the most important clinically and scientifically point “Smoking as a Risk Factor of Asthma” versus “Asthma as a risk factor of COPD Development”. This paragraph should describe potential differences in diagnosis, prognosis and treatment of four frequently wrongly used definitions:
A. A Asthma in a smoking patient
B. B Asthma /COPD overlap syndrome
C. C Asthma with persistent bronchial obstruction
D. D COPD in a patient with asthma history
I know that discussing points A to D will be difficult based on current literature but in my opinion the manuscript may be innovative only in case those points are addressed. This may end up in another chapter “area for future research”
4. The conclusions repeat several statements from previous manuscript parts. The conclusions should have been reduced to maximum of 2-3 rather short sentences.
I do hope that you will find my revision useful for increase your manuscript value. I’m looking forward for the opportunity to assess the revised manuscript.
Author Response
Reviewer 1
Dear Authors, I have read the text with interest especially because it is highly clinically relevant. I do agree with authors general approach that smoking is harmful for asthmatic patients and comprehensive efforts should be undertaken decrease smoking burden especially in asthmatic. Although I find the review well written and useful especially for young physicians there are some aspects which in my opinion are crucial before acceptance of the manuscript for publication.
Response
We would like to thank the reviewer for the kind remarks and the helpful recommendations to improve our manuscript.
- 1 Introduction
I would expect that the authors would describe in this part what was the methodology for review performance (systemic review/ random autonomic authors decision?). Which databases have been assessed? Which articles have been included or excluded, and what kea words have been used. In case the review was not systematic this should have been underlined in the introduction, but the method of literature search should be described.
Response
We have included a description of the methodology we employed for our search as requested in a separate paragraph titled methods.
“We performed a review of the literature across two databases (PubMed and Google Scholar), using relevant keywords: asthma and tobacco or smoking or smoke. We searched for systematic reviews, meta-analyses, observational studies and clinical trials that examined the effect of various exposures of tobacco in asthma. We included all the following types of studies: studies that examined smoking as a risk factor or a prognostic factor of asthma, studies that estimated the epidemiology of smoking in patients with asthma, studies that examined diagnostic and/ or therapeutic deviations in asthmatic smokers, studies that assessed the efficacy and/ or efficiency of smoking cessation options among asthma patients. We excluded studies that did not involve humans and were not published in English language. We did not enforce any limitation on the timing of publication.”
- 2 Diagnostic Challenges for Asthmatic Smokers: The authors explain most important challenges of differential diagnosis of asthma and COPD which includes Spirometry tests. Unfortunately dyspnea characteristics, which is weary different in asthmatic and COPD patients. This should be explained in detail with underlining nighttime dyspnea exacerbations are characteristic of asthma and exercise induced dyspnea in COPD.
Response
We have now mentioned the patterns of symptoms, and specifically dyspnea, among differences of asthma and COPD.
“The main symptoms of COPD are similar to the ones of asthma, but lack the characteristic variable pattern displayed in asthma.[8, 47] Specifically, dyspnea presents on exertion in early stages of COPD, while it manifests during exacerbations and nighttime in asthma.[48]”
- 3 I do luck a distinct chapter in which tips for differential diagnosis of “patients with asthma who smoke with present an accelerated rate of decline in lung function and persistent airflow obstruction due to airway remodeling with COPD which has developed in allergic patient. (the FEV1 decline and bronchial reactivity in both diseases should be addressed in larger detail.
Response
We have modified the chapter of diagnostic challenges to elucidate how differential diagnosis might be enabled for the two conditions, according to suggestions from both reviewers. We have also constructed a helpful diagnostic algorithm.
“The main symptoms that trigger a diagnostic examination for asthma are shortness of breath, cough, chest tightness and wheezing. Typically, patients have more than one of these symptoms, and symptoms have a certain pattern: they vary over time and in intensity, tend to worsen at night and early morning and may be triggered by certain exposures, such as tobacco and allergens.[1, 8] An important clinical diagnostic challenge is to distinguish between the two most prevalent airflow limitation disorders, asthma and COPD. COPD is a very common chronic respiratory condition, characterized by persistent symptoms and airflow limitation due to airway and/ or alveolar abnormalities. It has different pathophysiology than asthma, and is typically diagnosed in midlife or old age.[48, 49] The main symptoms of COPD are similar to the ones of asthma, but lack the characteristic variable pattern displayed in asthma.[8, 49] Specifically, dyspnea presents on exertion in early stages of COPD, while in asthma it is present during exposure to triggers, at night and/or during exacerbations and asthma patients are more likely to report wheezing and less likely to have chronic bronchitis symptoms.[50-52] Other attributes that might be helpful to differentiate asthma from COPD include sex, information from the patient’s family and personal history (family history of asthma, allergen sensitization, history of hay fever, eczema or allergic rhinitis, presence of comorbid diseases such as cardiovascular disease), reporting of symptoms and biomarkers (eosinophils, IgE, FeNO).[51, 52] A helpful diagnostic algorithm, adapted from the Greek Guidelines for COPD is pre-sented in Figure 1.[54] Last but not least, in early stages of COPD, clinical examination is normal, as is in asthma.[8, 49]
The diagnosis of asthma is made by lung function testing in patients with a clinically relevant presentation. Spirometry in asthma displays an expiratory airflow limitation and excessive variability of lung function.[1, 8] Existing literature has depicted that smoking initiation for the majority of people happens at a young age and the transition to regular smoking happens between adolescence and young adulthood.[53] Nicotine dependence develops very quickly in young people with symptoms of asthma, making quitting a difficult process.[54] Asthma patients that have regularly smoked since childhood or early adulthood, are more prone to have an impaired lung function and airway remodeling, ultimately causing fixed (non-reversible) airway obstruction.[55–59] This makes differentiation of asthma patients with a smoking history from COPD patients a strenuous task, given that smoking is the main risk factor of COPD and COPD patients have poorly reversible airflow limitation.[49] On top of that, certain COPD patients manifest airway hyper-responsiveness, which is the trademark feature of asthma.[60] However, the combination of greater lung function values, slower rates of lung function decline and marked airway hyperresponsiveness should steer the clinician towards an asthma diagnosis. [51, 52]”
4 Most importantly the Review in its current form is not addressing the most important clinically and scientifically point “Smoking as a Risk Factor of Asthma” versus “Asthma as a risk factor of COPD Development”. This paragraph should describe potential differences in diagnosis, prognosis and treatment of four frequently wrongly used definitions:
- A Asthma in a smoking patient
- B Asthma /COPD overlap syndrome
- C Asthma with persistent bronchial obstruction
- D COPD in a patient with asthma history
I know that discussing points A to D will be difficult based on current literature but in my opinion the manuscript may be innovative only in case those points are addressed. This may end up in another chapter “area for future research”
Response
As stated in the previous response, we have better clarified various phenotypes of patients. We also constructed a figure of a diagnostic algorithm that might help clinicians unveil distinct patterns.
We have also stated that asthma might be a risk factor of COPD and cited relevant literature.
“To complicate matters ever more, there is data supporting that asthma is a risk factor of COPD and that the association persisted even after adjusting for smoking.[23]”
- The conclusions repeat several statements from previous manuscript parts. The conclusions should have been reduced to maximum of 2-3 rather short sentences.
I do hope that you will find my revision useful for increase your manuscript value. I’m looking forward for the opportunity to assess the revised manuscript.
Response
We have modified and shortened the conclusions accordingly. We have also created a concise table of main findings.
“Asthma and smoking intertwine in various ways. Both active and passive smoking are commonly considered as risk factors of asthma, but evidence of a causal association is conflicting. Despite the well-known hazardous effects of tobacco in respiratory health, smoking is a quite common habit among asthma patients. Smoking affects airway morphology and alters inflammation of asthmatic individuals, worsening disease prognosis and making differential diagnosis of asthma and COPD rather laborious. Quitting smoking improves both lung function and symptoms of asthma patients, rendering imperative for their proper treatment to receive comprehensive smoking cessation interventions.”
Reviewer 2 Report
The authors have written a review entitled " Asthma and Tobacco Smoking." Authors’ efforts deserve recognition; however, the manuscript raises several questions and does not provide adequate data that need to be addressed.
1. Provide data about the prevalence of smoke in asthma. Is there geographical or national difference?
2. Please add a schematic figure showing the molecular mechanisms/effects of smoke on asthma.
3. This review discusses the role of smoke with details. However, there are several phenotypes of asthma (eosinophilic, neutrophilic…). Please add descriptions and compare the effects of smoke on different phenotypes of asthma.
4. Add definition and treatment of severe asthma and provide data of smoke on severe asthma.
5. Provide specific data/number on exacerbation rates and FEV1 change in asthma with smoke?
6. Indoor and outdoor air pollutant also affect asthma control. Discuss the combination effect of smoke and air pollutant on asthma.
7. Any diagnostic algorithm for the clinicians to differentiate asthma/copd with smoke?
8. There are limitations and pitfalls of previous studies of smoke on asthma. Please write a paragraph to summary suggestions for future research design. The main issues include phenotypes of asthma, dose/duration of cessation drugs, biomarker of oxidative stress and inflammation, ethnicity, ICS use, clinical outcomes.
9. Line 133, The definition of COPD is unclear. Review GOLD 2022 guideline for modification.
10. Line 278, expand the description of pharmacological treatments for smoking cessation, including cessation rates, side effects, pharmacological mechanism, drugs comparison.
11. A table of the impact of smoke on asthma would be nice
12. line 196, provide reference to smoke-related comorbidities in asthma
Author Response
Reviewer 2
The authors have written a review entitled " Asthma and Tobacco Smoking." Authors’ efforts deserve recognition; however, the manuscript raises several questions and does not provide adequate data that need to be addressed.
Response
We would like to thank the reviewer for the gracious input and the helpful suggestions to improve our manuscript.
- Provide data about the prevalence of smoke in asthma. Is there geographical or national difference?
Response
We mentioned the estimate metrics of smoking among asthma patients and cited relevant studies.
“Prevalence rate estimates of smoking range from 20% to 35% of asthma patients in published population wide studies, regardless of the country of origin of the sample used. The prevalence of active smoking in patients with asthma is therefore approximately the same as the prevalence in the general population.[18, 41–45]”
- Please add a schematic figure showing the molecular mechanisms/effects of smoke on asthma.
Response
We would like to refrain from this respectfully, given the fact that molecular mechanisms were beyond the scope of this review, and we focused on clinical findings from trials and epidemiologic studies.
- This review discusses the role of smoke with details. However, there are several phenotypes of asthma (eosinophilic, neutrophilic…). Please add descriptions and compare the effects of smoke on different phenotypes of asthma.
Response
We have elaborated on the absence of concrete evidence on this topic.
“The specific effects of smoking in distinct asthma phenotypes have not received much attention, so more research is needed to clarify the effect of smoking in different phenotypes of asthma. [18, 42]”
- Add definition and treatment of severe asthma and provide data of smoke on severe asthma.
Response
We have quoted the definition of severe asthma and elaborated on the potential contributory role of smoking on severe asthma.
“The realization that asthma constitutes a syndrome rather than a single disease has created a paradigm shift in its treatment in recent years.[9] This is mainly applied for patients on the severe spectrum of the disease, i.e. patients with poor symptom control and/ or frequent exacerbations while receiving maximal optimized controller therapy.[47, 105] For these patients, instead of enforcing a “one size fits all” strategy, therapy is personalized depending on disease phenotype and certain patient characteristics.[47] Considering what was discussed previously on the various adverse effects of tobacco on the prognosis of asthma, smoking cessation is strongly encouraged by latest guidelines on asthma management. What is more, tobacco smoking is nowadays considered among comorbid diseases, as a treatable trait that should be targeted to improve asthma control.[105, 106] Both Global Initiative of Asthma and joint European Respiratory Society/ American Thoracic Society guidelines cite smoking as a potential contributory factor of severe asthma, i.e. asthma that requires treatment with high doses of ICS plus a second controller to achieve disease control or that remains uncon-trolled.[107, 108]”
- Provide specific data/number on exacerbation rates and FEV1 change in asthma with smoke?
Response
We have specified the rate of exacerbations in current and former smokers.
“Specifically, more than half of current smokers had at least one exacerbation per year requiring systemic corticosteroids, compared to 40% of former smokers.[76]”
- Indoor and outdoor air pollutant also affect asthma control. Discuss the combination effect of smoke and air pollutant on asthma.
Response
We have elaborated on the above.
“The harmful effects of tobacco in asthma disease control might be aggravated in patients that are concurrently exposure to air pollution, either indoor or outdoor. These effects include asthma control, severity of the disease and lung function. Potential mechanisms of this effect include the exacerbation of inflammation and allergen-induced inflammation due to Th2 responses.”
- Any diagnostic algorithm for the clinicians to differentiate asthma/copd with smoke?
Response
We have modified the chapter of diagnostic challenges to elucidate how differential diagnosis might be enabled for the two conditions, according to suggestions from both reviewers. We have also constructed a diagnostic algorithm to help clinicians differentiate phenotypes.
“The main symptoms that trigger a diagnostic examination for asthma are shortness of breath, cough, chest tightness and wheezing. Typically, patients have more than one of these symptoms, and symptoms have a certain pattern: they vary over time and in intensity, tend to worsen at night and early morning and may be triggered by certain exposures, such as tobacco and allergens.[1, 8] An important clinical diagnostic challenge is to distinguish between the two most prevalent airflow limitation disorders, asthma and COPD. COPD is a very common chronic respiratory condition, characterized by persistent symptoms and airflow limitation due to airway and/ or alveolar abnormalities. It has different pathophysiology than asthma, and is typically diagnosed in midlife or old age.[48, 49] The main symptoms of COPD are similar to the ones of asthma, but lack the characteristic variable pattern displayed in asthma.[8, 49] Specifically, dyspnea presents on exertion in early stages of COPD, while in asthma it is present during exposure to triggers, at night and/or during exacerbations and asthma patients are more likely to report wheezing and less likely to have chronic bronchitis symptoms.[50-52] Other attributes that might be helpful to differentiate asthma from COPD include sex, information from the patient’s family and personal history (family history of asthma, allergen sensitization, history of hay fever, eczema or allergic rhinitis, presence of comorbid diseases such as cardiovascular disease), reporting of symptoms and biomarkers (eosinophils, IgE, FeNO).[51, 52] A helpful diagnostic algorithm, adapted from the Greek Guidelines for COPD is pre-sented in Figure 1.[54] Last but not least, in early stages of COPD, clinical examination is normal, as is in asthma.[8, 49]
The diagnosis of asthma is made by lung function testing in patients with a clinically relevant presentation. Spirometry in asthma displays an expiratory airflow limitation and excessive variability of lung function.[1, 8] Existing literature has depicted that smoking initiation for the majority of people happens at a young age and the transition to regular smoking happens between adolescence and young adulthood.[53] Nicotine dependence develops very quickly in young people with symptoms of asthma, making quitting a difficult process.[54] Asthma patients that have regularly smoked since childhood or early adulthood, are more prone to have an impaired lung function and airway remodeling, ultimately causing fixed (non-reversible) airway obstruction.[55–59] This makes differentiation of asthma patients with a smoking history from COPD patients a strenuous task, given that smoking is the main risk factor of COPD and COPD patients have poorly reversible airflow limitation.[49] On top of that, certain COPD patients manifest airway hyper-responsiveness, which is the trademark feature of asthma.[60] However, the combination of greater lung function values, slower rates of lung function decline and marked airway hyper-responsiveness should steer the clinician towards an asthma diagnosis. [51, 52]”
- There are limitations and pitfalls of previous studies of smoke on asthma. Please write a paragraph to summary suggestions for future research design. The main issues include phenotypes of asthma, dose/duration of cessation drugs, biomarker of oxidative stress and inflammation, ethnicity, ICS use, clinical outcomes.
Response
We have stressed the need for additional research on these topics.
“The specific effects of smoking in distinct asthma phenotypes has not received much attention, so more research is needed to clarify the effect of smoking in different phe-notypes of asthma. [18, 42]”
“The number of clinical trials examining the efficacy of various smoking cessation strategies and medication exclusively in asthma patients is very limited.[120, 123–125] More clinical trials should be designed to assess and compare the effectiveness of ex-isting smoking cessation interventions in asthma patients. Existing guidelines for asthma do not make any specific recommendations for smoking cessation interven-tions (i.e. specific therapeutic options and elaborate regimens or doses), while they emphasize that quitting should be encouraged and cessation support should be offered for all patients.[18, 75]”
“More pragmatic clinical trials, with less strict exclusion criteria, should be conducted to elucidate the comparative efficacy of various medications and treatment strategies in asthma patients.[18, 42, 94]”
- Line 133, The definition of COPD is unclear. Review GOLD 2022 guideline for modification.
Response
We have now included the latest COPD definition, along with the reference, as recommended.
“An important clinical diagnostic challenge is to distinguish between the two most prevalent airflow limitation disorders, asthma and COPD. COPD is a very common chronic respiratory condition, characterized by persistent symptoms and airflow limitation due to airway and/ or alveolar abnormalities. It has different pathophysiology than asthma, and is typically diagnosed in midlife or old age.[48, 49]”
- Line 278, expand the description of pharmacological treatments for smoking cessation, including cessation rates, side effects, pharmacological mechanism, drugs comparison.
Response
We have further elaborated on potential smoking cessation options as requested.
“There is consensus that the most efficient smoking cessation intervention is comprehensive treatment, combining behavioral counseling, pharmacotherapy, and follow-up support.[4, 119] Currently there are seven effective pharmacological treatments for smoking cessation approved by the US Food and Drug Administration. These include varenicline, bupropion and five forms of nicotine replacement therapy.[4, 119] The efficacy for abstinence for these options, when compared to placebo, are estimated at RR 2.24 (95% CI, 2.06 – 2.43) for varenicline, 1.64 (95% CI, 1.52 – 1.77) for bupropi-on, 1.49 (95% CI, 1.40 – 1.60) for gum, 1.64 (95% CI, 1.53 – 1.75) for patch, 1.52 (95% CI, 1.32 – 1.74) for lozenge, 1.90 (95% CI, 1.36 – 2.67) for inhaler, 2.02 (95% CI, 1.49 – 2.73) for nasal spray. The most common side effect for each of the above medications are: nausea for varenicline, insomnia for bupropion, jaw pain for gum, skin reactions for patch, hiccups for lozenge, cough for inhaler and nasal irritation for nasal spray.[4, 119]”
- A table of the impact of smoke on asthma would be nice
Response
We have constructed a table with main popular opinion beliefs on the association of smoking with asthma and what real evidence suggests, as a summary of the findings of our review. The form enables rapid reader viewing of main findings and easily conveys the information to non-specialized readers.
- line 196, provide reference to smoke-related comorbidities in asthma
Response
We have included relevant references for this information.
“Furthermore, asthmatic smokers display an enlarged prevalence of various comorbid conditions according to published studies. These conditions include COPD, perennial rhinitis, seasonal rhinitis, lung cancer, coronary heart disease, arrhythmias, hypertension, diabetes mellitus, osteoporosis and prostate hyperplasia.[76, 77]”
Round 2
Reviewer 1 Report
Congratulations!
In my opinion the manuscript is suitable for publication in its current form.
Best Regards,
Reviewer 2 Report
The authours answered the revisions with good and detalied corrections.